# Calculating the Effective Center Wavelength for Heterodyne Interferometry of an Optical Frequency Comb

**Shilin Xiong [1], Yue Wang [1], Yawen Cai [2], Jiuli Liu [2], Jie Liu [2] and Guanhao Wu [1,\*]**

[1] State Key Laboratory of Precision Measurement Technology and Instruments, Department of Precision Instrument, Tsinghua University, Beijing 100084, China; xiongshilin0808@163.com (S.X.); wy310275@163.com (Y.W.)

[2] Institute of Spacecraft System Engineering, China Academy of Space Technology, Beijing 100094, China; ywcai02@gmail.com (Y.C.); justin_2009@163.com (J.L.); liujie@cast.cn (J.L.)

[\*] Correspondence: guanhaowu@mail.tsinghua.edu.cn; Tel.: +86-010-6279-8763

**Abstract:** Heterodyne interferometry based on an optical frequency comb (OFC) is a powerful tool for distance measurement. In this paper, a method to calculate the effective center wavelength of wide spectrum heterodyne interference signal was explored though both simulation and experiment. Results showed that the effective center wavelength is a function of the spectra of the two interfered beams and time-delay of the two overlapped pulses. If the product of the spectra from two arms is symmetric, the effective center wavelength does not change with time-delay of the two pulses. The relative difference between the simulation and experiment was less than 0.06%.

**Keywords:** optical frequency comb; heterodyne interferometry; center wavelength

## 1. Introduction

An optical frequency comb (OFC) emits an evenly spaced ultra-short pulse train with a broad spectrum consisting of discrete, narrow lines with uniform mode-spacing [1,2]. The absolute frequency of each mode can be expressed as:

$$f_m = m f_{\text{rep}} + f_{\text{ceo}}, \tag{1}$$

where $f_{\text{rep}}$ is the repetition rate, $f_{\text{ceo}}$ stands for the carrier-envelope-offset frequency, and $m$ is the mode order. When $f_{\text{rep}}$ and $f_{\text{ceo}}$ are stabilized referencing a frequency standard, OFC becomes an ultra-precise ruler in the space, time, and frequency domain [3–5]. Therefore, it is useful for absolute distance measurement.

In the past decade, numerous methods based on OFC have been proposed to measure absolute distance with high precision. These methods can be categorized into several groups according to the measurement principle and include: using the inter-mode beat signals of the comb [6,7], applying dispersive interferometry [8–13], using the pulse separation distance as a ruler [14–19], and the dual-comb method [20–25]. In order to suppress the effect of intensity noise, heterodyne interferometry has been introduced into the OFC distance measurement system [26]. The OFC heterodyne interferometry displays excellent results in temporal coherence interferometry, two-color correction of the refractive index of air [27,28], and pulse-to-pulse alignment [29]. Therefore, it is a powerful tool for distance measurement.

In a traditional laser heterodyne interferometer, the accuracy of the wavelength is very important for the distance measurement. However, in an OFC heterodyne interferometer, the light source has a wide optical spectrum; the spectral width of an optical frequency comb is normally tens of nanometers.

Additionally, the optical spectra of the beams in the two arms of the interferometer can be different, and thus, it is ambiguous to determine the center wavelength. Determining the effective center wavelength of the interference signal is a basic but essential question for the OFC heterodyne interferometry which has not yet been investigated in detail.

In order to resolve this problem, this paper introduced a method for calculating the effective center wavelength of the OFC heterodyne interference signal in an equal-arm interferometer included in a commercial interferometer. The method was then verified by simulation and experimental results.

## 2. Methods and Experiments

A schematic of the OFC heterodyne interferometer used in this study is provided in Figure 1. The light source (OFC) was a homemade mode-locked erbium-doped fiber femtosecond laser. The central wavelength of the OFC was 1560 nm, the full width at half maximum (FWHM) of the spectrum was 55 nm, the output power was approximately 8 mW, and the repetition frequency was stabilized to a frequency synthesizer (78 MHz, 33250A, Agilent, Santa Clara, CA, USA), which was referenced to an atomic clock (5071A, Symmetricom, San Jose, CA, USA).

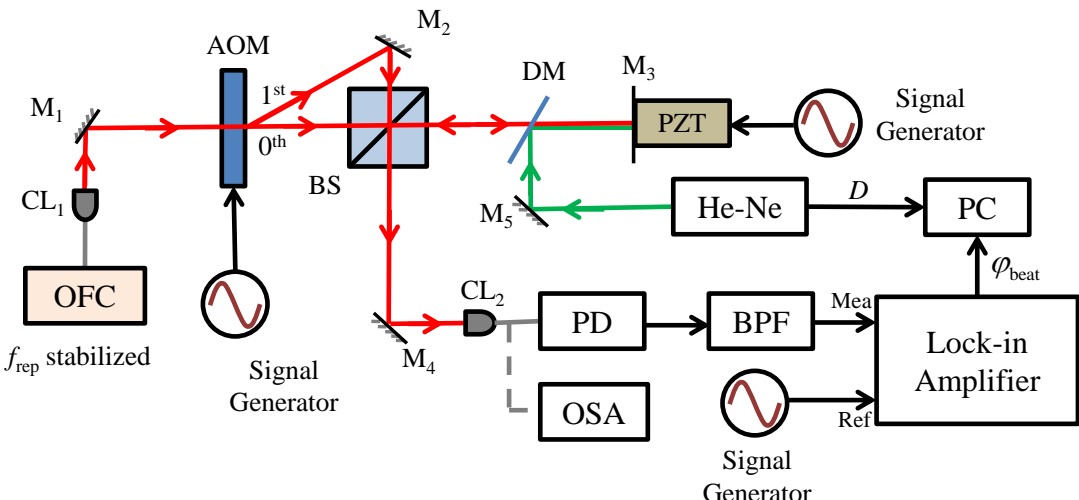

**Figure 1.** Schematic of the optical frequency comb (OFC) heterodyne interference system. Gray lines: optical fibers; red and green lines: optical paths in free space; black lines: electrical connections. OFC: optical frequency comb; CL: collimating lens; M: mirror; AOM: acoustic optical modulator; BS: beam splitter; DM: dichroic mirror; PZT: piezo-electric transducer; OSA: optical spectrum analyzer; PD: photodetector; BPF: bandpass filter; He-Ne: commercial interferometer; PC: computer.

The output laser beam from the OFC passed through an acousto-optic modulator (AOM, MGAS80-A1, AA Opto Electronic, Orsay, France) driven by a sinusoidal signal at a constant frequency $f_{AOM}$ = 80 MHz. After the AOM, the zero-order beam travels along the original direction with the original frequency, while the first-order beam spreads in another direction due to optical diffraction, and its frequency is shifted by $f_{AOM}$. The zero-order beam travels through a beam splitter (BS) and arrives at mirror $M_3$, then reflects back to BS and overlaps with the first-order beam which is adjusted by mirror $M_2$. Two beams were coupled into an optical fiber together by a collimating lens ($CL_2$). The optical lengths of the two beams were set to be equal. The optical spectra of the two beams were then individually measured by an optical spectrum analyzer (OSA, AQ6370C, Yokogawa Electric, Musashino, Tokyo, Japan). The first-order beam is referred to as the reference arm, while the zero-order beam is called the measurement arm. Figure 2a illustrates the optical spectrum of the two arms. Note that the optical spectra of the two beams are different, due to the diffraction of the AOM.

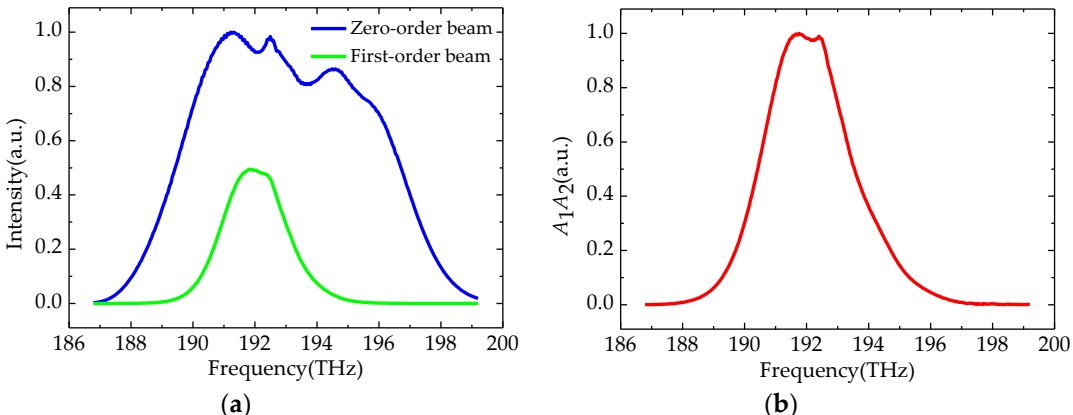

**Figure 2.** (**a**) Optical spectrum of two beams; (**b**) Product of electric amplitude of two arms.

Because the frequency of the first-order beam was shifted by $f_{\text{AOM}}$, a heterodyne interference signal was detected by the photodetector (PD, Model 1811, New Focus, CA, USA), and the frequency of the heterodyne interference signal was $f_{\text{beat}} = f_{\text{AOM}} - f_{\text{rep}} = 2$ MHz. The beat signal was extracted by a bandpass filter and then sent into a lock-in amplifier (SR844, Stanford Research System, Sunnyvale, CA, USA) to measure the phase compared with the reference signal.

The beam of the commercial interferometer overlapped with the beam of OFC at the dichroic mirror (DM). The target mirror $M_3$ was driven by a piezo-electric transducer (PZT). The OFC heterodyne interferometer and the commercial interferometer were both used to measure the displacement of $M_3$ simultaneously, see Section 3 for details. According to the comparison results, the effective center wavelength of the OFC heterodyne interference signal can then be estimated.

In the following section, a theoretical method to calculate the effective center wavelength of the OFC heterodyne interferometer was introduced. The complex amplitude of the electric field of two beams can be expressed as:

$$E_1(t) = \sum_m A_1 e^{i[2\pi f_m(t-t_1)+\beta_m]}, \tag{2}$$

$$E_2(t) = \sum_m A_2 e^{i[2\pi(f_m+f_{beat})(t-t_2)+\beta_m]}, \tag{3}$$

where $A_1(f_m) \propto \sqrt{P_1(f_m)}$, $A_2(f_m) \propto \sqrt{P_2(f_m)}$, $P_1$, and $P_2$ are the optical spectral intensity of the two beams as shown in Figure 2a. The imaginary unit is $i$, $t_1$, and $t_2$ are the time delay of the two beams, and $\beta_m$ is the initial phase of the $m^{\text{th}}$ mode. The interference signal intensity is:

$$I = |E_1 + E_2|^2 = (E_1 + E_2) \cdot (E_1 + E_2)^*. \tag{4}$$

After a bandpass filter, the beat signal of OFC heterodyne interferometry is:

$$I_{\text{beat}} = \sum_m 2A_1 A_2 \cos[2\pi f_{beat}(t - t_2) + 2\pi f_m(t_1 - t_2)]. \tag{5}$$

Equation (5) is equivalent to this form:

$$I_{\text{beat}} = 2a \cos[2\pi f_{\text{beat}}(t - t_2) + \varphi_{\text{beat}}], \tag{6}$$

where $\varphi_{\text{beat}}$ is the phase measured by the lock-in amplifier, and:

$$a \cos \varphi_{\text{beat}} = \sum_m A_1(f_m) A_2(f_m) \cos 2\pi f_m \tau, \tag{7}$$

$$a \sin \varphi_{\text{beat}} = \sum_m A_1(f_m) A_2(f_m) \sin 2\pi f_m \tau, \tag{8}$$

where $\tau = t_1 - t_2$ is the relative time delay of the two beams. According to Equations (7) and (8):

$$\tan \varphi_{\text{beat}} = \frac{\sum\limits_m A_1(f_m) A_2(f_m) \sin 2\pi f_m \tau}{\sum\limits_m A_1(f_m) A_2(f_m) \cos 2\pi f_m \tau}, \tag{9}$$

then $\varphi_{\text{beat}}$ is a function of relative time delay $\tau$:

$$\varphi_{\text{beat}} = \varphi(\tau). \tag{10}$$

For the continuous-wave laser heterodyne interferometer, $\varphi(\tau) = 2\pi f_c \tau$, thus, the effective center frequency for OFC heterodyne interferometry is:

$$f_c = \frac{1}{2\pi} \varphi'(\tau). \tag{11}$$

The effective center wavelength can then be calculated using $\lambda_c = c/f_c$, where $c$ is the velocity of light in the vacuum.

Figure 2b illustrates the product of the electric amplitude of the two beams, which is the function of the optical frequency. If the product is symmetrical for the frequency at $f_{m0}$, Equation (9) can be simplified as:

$$\tan \varphi_{\text{beat}} = \frac{\sin 2\pi f_{m0} \tau}{\cos 2\pi f_{m0} \tau} = \tan 2\pi f_{m0} \tau, \tag{12}$$

therefore, $\varphi_{\text{beat}} = 2\pi f_{m0} \tau$, meaning that the effective center wavelength does not change with the relative time delay of two beams. In this study, however, the product was not symmetric, thus, the effective center wavelength was calculated by simulation.

In this simulation, the parameters of the system were the same as the experimental setup, including the electric amplitude of the two beams and repetition frequency of OFC. The carrier-envelope-offset frequency was regarded as zero. The phase $\varphi_{\text{beat}}$ of the heterodyne interference signal was calculated at different relative time delay according to Equation (9), and numerical differentiation methods from Equations (10) and (11) were used to calculate the effective center wavelength at different relative time delay.

## 3. Results

To calculate the effective center wavelength of the OFC heterodyne interferometer, a square wave signal generated from a signal generator controlled the displacements of the PZT. The frequency of the square wave signal is 10 Hz, with a peak-to-peak value of 0.4 V. The sensitivity of PZT to light path variations was about 6.4 μm/V, and the positions of $M_3$ were monitored precisely by the commercial interferometer.

Figure 3a shows the phase change of OFC heterodyne interferometer and the commercial interferometers while tuning the displacements of PZT.

The effective center wavelength at this position is calculated by:

$$\lambda_c = \frac{2\pi n \Delta D}{\Delta \varphi}, \tag{13}$$

where $n$ is the refractive index of air. Since the displacement was larger than the wavelength, it was necessary to unwrap the phase when calculating the effective center wavelength. Environmental parameters were recorded throughout the experiment in order to calculate the air refractive index for both commercial interferometer and OFC heterodyne interferometer.

To evaluate the stability of the effective center wavelength while the relative position of two overlapped pulses change, the bias voltage of the square wave was altered to shift the equilibrium position of PZT. The bias voltage was changed by six steps at 1 V interval (corresponding to 6.4 μm).

At each position, the effective center wavelength was measured 10 times. The average value and standard deviation are presented in Figure 4. The simulation results are also provided for comparison.

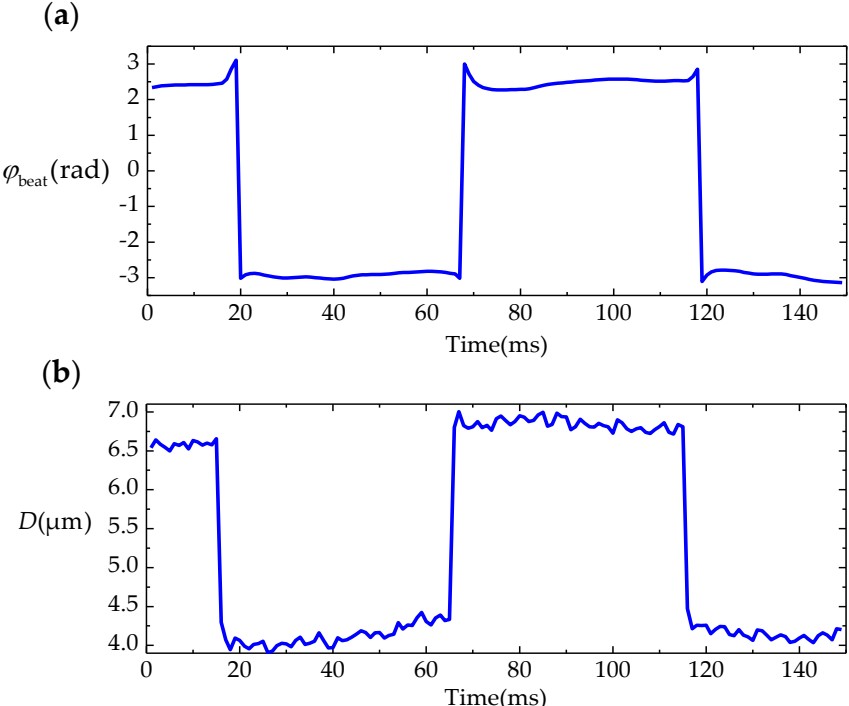

**Figure 3.** (**a**) $\varphi_{\text{beat}}$ change with time measured by lock-in amplifier; (**b**) displacement change with time measured by commercial interferometer.

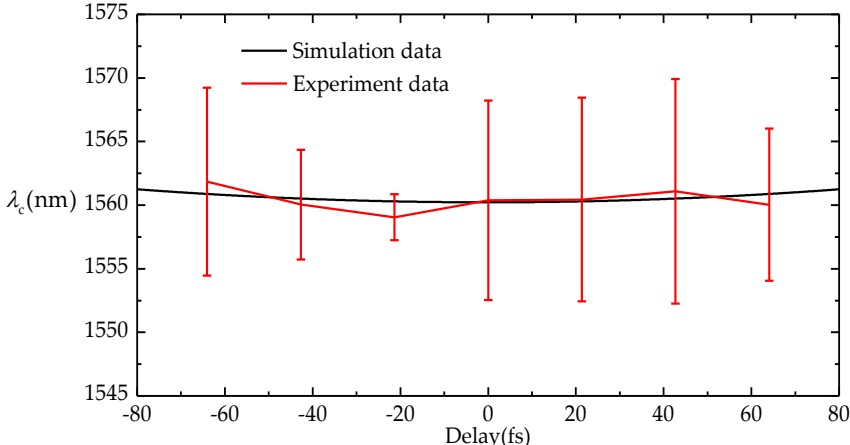

**Figure 4.** Effective center wavelength $\lambda_c$ with different pulse delay. The data was measured by scanning the mirror $M_3$ through modulating the voltage of PZT. Black line: results of simulation; red line: results of experiments. Error bars represents $+/-$ σ (standard deviation) for 10 measurements.

When the pulse delay changed from $-80$ fs to 80 fs, the effective center wavelength changed approximately 1.02 nm according to the simulation results. This is because the product of the electric amplitude of two beams was not symmetric, and the function of phase $\varphi_{\text{beat}}$ change over the relative time delay was not a linear function. The biggest relative difference between the simulation and experiment was approximate to 0.06%, which was predominantly caused by the random error of the commercial interferometer and the phase drifting caused by the AOM. The optical frequency comb had an especially wide spectrum; its coherence length was very short. When calculating the effective center wavelength for heterodyne interferometry, the displacement was only about 2.5 μm. The repeatability

of commercial interferometer was only about 15 nm for the short displacement measurement, which was the main error source in the evaluation results shown in Figure 4. Note that the two beams in this experiment come out of the same optical fiber, thus, the initial phase $\beta_m$ of two beams was equal. However, if the beams would have come from different optical fiber, the initial phase $\beta_m$ of the two beams would be different because of the chirp of optical fiber. This case requires further investigation for calculating the effective center wavelength.

## 4. Conclusions

A method was proposed to calculate the effective center wavelength for an OFC heterodyne interferometer based on the spectra of two interfered beams. An experimental setup was established to verify the theoretical formula. The results show that the theoretical analysis corresponds well with the experimental results, and illustrate that the proposed model is reasonable and effective. This method is an important tool for OFC heterodyne interferometry that can be used for ranging or pulse alignment and other applications.

**Author Contributions:** G.W. and S.X. conceived and designed the experiments; Y.C., J.L. (Jiuli Liu), and J.L. (Jie Liu) discussed the experimental scheme; S.X. and Y.W. performed the experiments; S.X. analyzed the data; S.X. and G.W. wrote the paper. All authors have read and approved the final manuscript.

**Funding:** This research was funded by the National Key Research and Development Project (2016YFF0101804), the National Natural Science Foundation of China (61575105, 61611140125), the Beijing Natural Science Foundation (3182011), and Shenzhen fundamental research funding (Grant No. JCYJ20170412171535171).

**Conflicts of Interest:** The authors declare no conflict of interest.

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
