# Peer review of "Calculating the Effective Center Wavelength for Heterodyne Interferometry of an Optical Frequency Comb"

_applsci, doi:10.3390/app8122465_

Reviewer 1 Report

This paper presents a method to evaluate effective center wavelength for heterodyne interferometry using an optical frequency comb.The manuscript is well prepared, and sounds scientifically rigorous.

Meanwhile, due to the lack of information on the target value (uncertainty) for the evaluation of effective center wavelength, the reviewer cannot judge the advantage of this method compared with the conventional methods. Especially, the standard deviation of plus/minus 10 nm for the wavelengh measurement sounds too much.

Furthermore, in this study, a comparison has been made by using a commercial CW laser (He-Ne). The authors are expected to carry out a detailed analysis on the measurement uncertainty of the proposed method to verify its feasibility for evaluation of the effective center wavelength measurement. At least, the authors are expected to discuss possible root causes of the poor standard deviation (repeatability) of the experimental results at each time delay.

Reviewer 2 Report

This paper introduces a method for calculating the effective centre wavelength for a heterodyne interferometer based on an optical frequency comb. The paper is well-written and clear, although some minor improvements to the English are required.

I found that mentioning the commercial interferometer in line 83, but only explaining how it was used around line ~130, was a little confusing. A comment like "See section 3 for details" in the sentence in line 83 would be helpful.

In lines 40-42, the authors state that "In a traditional laser heterodyne interferometer, the accuracy of the wavelength is very important for the distance measurement. However, in an OFC heterodyne interferometer, the light source has a wide optical spectrum." Can any quantatitive information be given here? What effect does lack accuracy of the effective centre wavelength have on distance measurements with OFC heterodyne interferometry?

In the conclusion, it would be useful if the authors could comment on the implications of their work, e.g. by explaining in more detail why it is an important tool for OFC heterodyne interferometry.

There were a few typos etc, particularly on the first page (changes in capitals)

Line 13 - "based on AN optical frequency comb"

Line 16 - "is a function of THE spectra"

Line 17 - "if the product of THE SPECTRA from two arms"

Line 23 - "AN evenly spaced ultra-short pulse train"

Line 24 - May be clearer to change to "consisting of discrete, narrow lines with uniform mode-spacing"?
